# AMPHIBIAN: A META-LEARNER FOR REHEARSAL-FREE FAST ONLINE CONTINUAL LEARNING

## ABSTRACT

Online continual learning is challenging as it requires fast adaptation over a stream of data in a non-stationary environment without forgetting the knowledge acquired in the past. To address this challenge, in this paper, we introduce Amphibian - a gradient-based meta-learner that learns to scale the direction of gradient descent to achieve the desired balance between fast learning and continual learning. For this purpose, using only the current batch of data, Amphibian minimizes a meta-objective that encourages alignments of gradients among given data samples along selected basis directions in the gradient space. From this objective, it learns a diagonal scale matrix in each layer that accumulates the history of such gradient alignments. Using these scale matrices Amphibian updates the model online only in the directions having positive cumulative gradient alignments among the data observed for far. With evaluation on standard continual image classification benchmarks, we show that such meta-learned scaled gradient descent in Amphibian achieves state-of-the-art accuracy in online continual learning while enabling fast learning with less data and few-shot knowledge transfer to tasks. Finally, with loss landscape visualizations, we show such gradient updates incur minimum loss to the old task enabling fast continual learning in Amphibian.

## 1 INTRODUCTION

Autonomous intelligent systems are envisioned to operate in non-stationary environments where distribution of online data streams changes over time. In such environments, AI models (usually artificial neural networks, ANNs) need to acquire knowledge quickly while maintaining the stability of past experiences. This is a challenging scenario as the learning method needs to strike the right balance between *learning without forgetting* and *fast learning* objectives. However, standard gradient-based training methods for ANNs overwrite the past knowledge with the information from the new batch of data - leading to 'catastrophic forgetting' (Mccloskey & Cohen, 1989). Such forgetting prevents effective knowledge transfer from the past thus also hampering fast learning ability.

To address these challenges, a popular line of work in continual learning (CL) (Ring, 1998; Hadsell et al., 2020) uses memory rehearsal (Robins, 1995; Chaudhry et al., 2019b; Lopez-Paz & Ranzato, 2017) - where a subset of past data is stored in a memory buffer and used with the current batch of data to jointly train the model. Such rehearsal-based strategy guides the optimization process such that losses of the past data do not increase, preventing catastrophic forgetting. However, effectiveness of these methods depends on large memory storage which also arises data privacy concerns. In contrast, rehearsal-free methods (Kirkpatrick et al., 2017; Zenke et al., 2017; Saha et al., 2021b; Serrà et al., 2018) in continual learning use explicitly designed regularization objectives and/or constrained gradient update rules to prevent forgetting. Though these methods are effective in offline (multi-epoch) CL setups, compared to rehearsal-based methods they underperform in online continual learning (OCL) (Mai et al., 2022). This is primarily due to the added objective or constraints that focus on forgetting mitigation rather than encouraging fast learning. From fast learning viewpoint, meta-learning (Finn et al., 2017) or 'learning to learn' (Thrun & Pratt, 2012) is an exciting proposition since it optimizes a meta-objective that encourages representation learning in ANNs suitable for fast adaptation. Such meta-objective is adapted in Javed & White (2019); Beaulieu et al. (2020); Caccia et al. (2020) for pre-training models offline, then deployed for continual learning tasks. In contrast, Riemer et al. (2019); Gupta et al. (2020) adapted the meta-objective for fully online continual learning. However, they use memory rehearsal to mitigate forgetting.

In this paper, we propose a meta-learner - **Amphibian** - that learns fast with minimum forgetting during online continual learning. Without any memory rehearsal, Amphibian achieves better balance between *learning without forgetting* and *fast learning* with three key components. **First**, to obtain meta-gradients, it optimizes a meta-objective that on top of minimizing loss on the given batch of samples, encourages their gradient alignments along the selected basis directions representing the gradient space. **Second**, from the same meta-objective, it *learns* diagonal scaling matrices that contain learning/scaling rates along each gradient basis. In our formulation, each scale value accumulates the history of gradient alignments (along the corresponding basis) among the observed samples over the entire learning sequence. **Finally**, it scales the meta-gradients with scaling matrices to update the model along the directions with cumulative positive gradient alignments among the observed data. Thus, the combination of meta-objective optimization and meta-learned gradient scaling enables Amphibian to learn fast and continually. We evaluate Amphibian in various online continual learning setups (Gupta et al., 2020; Shim et al., 2021) on long and diverse sequences of image classification tasks (including ImageNet-100) using different network architectures (including ResNet) and achieve better performance in both continual and fast learning metrics compared to the twelve most relevant baselines. We summarize the **contributions** of this paper as follows:

- We introduce Amphibian which minimizes a novel meta-objective and uses meta-learned gradient scaling to enable fast online continual learning without rehearsal.

- With evaluation on long sequences of tasks, we show that Amphibian not only learns continually with SOTA accuracy but also demonstrates the ability of a truly fast learner by learning fast with less data and enabling few-shot knowledge transfer to the new tasks.

- We analyze a regularized version of Amphibian - Amphibian-$\beta$ and provide insight that regularized objectives or constraints used by the representative rehearsal-free methods to minimize forgetting restrict the fast learning ability of the model in OCL.

- With visualization of loss landscapes of sequential tasks, we show that scaled model update in Amphibian along gradient directions with positive cumulative gradient alignments induces minimum to no increase in loss of the past tasks which enables continual learning.

## 2 RELATED WORKS

**Online Continual Learning (OCL).** We consider a supervised learning setup where $T$ tasks $[\tau_1, \tau_2, ..\tau_T]$ are learnt by observing their training data $[\mathcal{D}^1, \mathcal{D}^2, ..\mathcal{D}^T]$ sequentially. At any time-step $j$, the learning model receives a batch of data, $\mathcal{B}_i^j$ or simply $\mathcal{B}_i = \{(x_n^j, y_n^j)\}_{n=0}^{N_i}$ as the set of $N_i$ input-label pairs randomly drawn from the current data stream, $\mathcal{D}_i$. In **online** continual learning (Mai et al., 2022), the model needs to learn from a single pass over these data streams with the objective of minimizing the empirical risk on the data from all the $t$ tasks seen so far. The objective (Gupta et al., 2020; Verwimp et al., 2021) is given by:

$$\sum_{i=1}^{t} \mathbb{E}_{\mathcal{B}_i}[\ell_i(\theta; \mathcal{B}_i)] = \mathbb{E}_{\mathcal{B}_{1:t}} \sum_{i=1}^{t} [\ell_i(\theta; \mathcal{B}_i)]. \tag{1}$$

Here $\ell_i(.;.)$ is the loss function to be minimized for task $\tau_i$ by updating the model parameters $\theta$.

**Rehearsal-based Methods.** The above risk minimization requires all the data, $\mathcal{B}_{1:t-1}$ from past tasks which may not be accessible to OCL agents at the same time. Rehearsal or experience replay (ER) (Robins, 1995; Chaudhry et al., 2019b) methods offer a solution by storing a limited amount of past data in episodic memory, $\mathcal{D}^{\mathcal{M}}$. Such techniques then sample a memory batch, $\mathcal{B}_{1:t-1}^{\mathcal{M}} \sim \mathcal{D}^{\mathcal{M}}$ (that approximates $\mathcal{B}_{1:t-1}$) for jointly minimizing the objective in Equation 1 with the current batch, $\mathcal{B}_t$. Later works built on this idea where they differ in the way memory is selected and replayed. For instance: GSS (Aljundi et al., 2019b) selects memory based on gradients, MIR (Aljundi et al., 2019a) selects memory that incurs maximum change in loss, ASER Shim et al. (2021) performs memory selection and retrieval using Shapley value scores. To improve replay, RAR Zhang et al. (2022) uses repeated augmented rehearsal, CLS-ER (Arani et al., 2022) proposes dual memory learning and DER Buzzega et al. (2020) uses logit distillation loss. Gradient Episodic Memory (GEM) (Lopez-Paz & Ranzato, 2017) and Averaged-GEM (A-GEM) (Chaudhry et al., 2019a) use memory data to compute gradient constraints for new task so that loss on past the tasks does not increase.

**Rehearsal-free Methods.** One line of work in this category expands the network (Rusu et al., 2016; Yoon et al., 2018) for continual learning but they are not evaluated for OCL. Other methods

under this category minimize the loss ($\mathbb{E}_{\mathcal{B}_t}[l_t(\theta; \mathcal{B}_t)]$) on the current batch with additional regularization terms and/or constraints on gradient updates. For example, Elastic Weight Consolidation (EWC) (Kirkpatrick et al., 2017) and Synaptic Intelligence (SI) (Zenke et al., 2017) add quadratic regularization terms that penalize changes in the important parameters for the past task. While EWC computes parametric importance from the Fisher diagonal matrix after training, SI finds them online during training from the loss sensitivity of the parameters. Gradient projection methods (Zeng et al., 2019; Farajtabar et al., 2020; Saha et al., 2021b; Saha & Roy, 2023) constrain the current gradient, $\nabla_\theta l_t$ to be orthogonal to the past gradient directions. For instance, Gradient Projection Memory (GPM) (Saha et al., 2021b) optimizes a new task in the orthogonal directions to the gradient spaces important for the past tasks, whereas Scaled Gradient Projection (SGP) (Saha & Roy, 2023) relaxes the constraint in GPM to allow gradient updates along the old gradient spaces. Natural Continual Learning (NCL) (Kao et al., 2021) learns continually by combining gradient projection with regularization. Our proposed Amphibian is also a rehearsal-free method. However, unlike these methods it does not need any explicit regularization or constraints; rather it *meta-learns from data to scale the gradients* for fast continual learning. In the next section, we introduce a meta-learning method - MAML (Finn et al., 2017) and then discuss relevant meta-learning-based continual learning works.

## 3 PRELIMINARIES

**Model-Agnostic Meta-Learning (MAML).** MAML (Finn et al., 2017) is a widely used gradient-based meta-learner that utilizes bi-level (inner-loop and outer-loop) optimization to obtain model parameters, $\theta_0$ that is amenable to fast adaptation to new tasks. MAML trains the model on a set of $T$ tasks simultaneously where each task, $\tau_i$ has a dataset, $\mathcal{D}^{\tau_i} = \{\mathcal{D}_{in}^{\tau_i}, \mathcal{D}_{out}^{\tau_i}\}$ partitioned for inner and outer loop optimization. One step of **inner-loop optimization** on task $\tau_i$ is defined as:

$$U(\theta_0^{\tau_i}; \mathcal{D}_{in}^{\tau_i}) = \theta_1^{\tau_i} = \theta_0^{\tau_i} - \alpha \nabla_{\theta_0^{\tau_i}} \ell_{in}(\theta_0^{\tau_i}; \mathcal{D}_{in}^{\tau_i}), \tag{2}$$

where $U(.;.)$ is a stochastic gradient descent (SGD) operator, $\alpha$ is learning rate and $\ell_{in}$ is inner-loop loss function. $U$ can be composed for $k$ such updates as $U_k(\theta_0^{\tau_i}; \mathcal{D}_{in}^{\tau_i}) = U.. \circ U \circ U(\theta_0^{\tau_i}; \mathcal{D}_{in}^{\tau_i}) = \theta_k^{\tau_i}$. In the **outer-loop optimization**, loss for each task is computed at corresponding, $\theta_k^{\tau_i}$ on $\mathcal{D}_{out}^{\tau_i}$ and expected loss over all the tasks (meta-loss) is minimized (Equation 3) to obtain the parameters, $\theta_0$.

$$\min_{\theta_0} \mathbb{E}_{\tau_{1:t}}[\ell_{out}(\theta_k^{\tau_i}; \mathcal{D}_{out}^{\tau_i})] \tag{3}$$

**Meta-learning and Continual Learning.** The above meta-loss minimization trains a model for fast adaptation, however it does not explicitly encourage continual learning. Thus, Javed & White (2019) proposed online-aware meta-learning where at first, a model is pre-trained offline on a set of tasks to learn a better representation for CL, then keeping that representation frozen, the rest of the network is fine-tuned on CL tasks. The authors in Beaulieu et al. (2020); Lee et al. (2021); Caccia et al. (2020) used such meta-learning-based offline pre-training strategy, while allowing varying degrees of adaptation to the model during CL tasks. In contrast, Meta Experience Replay (MER) (Riemer et al., 2019) combines meta-objective of Reptile (Nichol et al., 2018) with memory rehearsal for OCL, whereas La-MAML (Gupta et al., 2020) minimizes the MAML objective (Equation 3) in online setup where losses on the past tasks are computed on the memory batch, $\mathcal{B}_{1:t-1}^{\mathcal{M}} \sim \mathcal{D}^{\mathcal{M}}$. Unlike these methods, we train a model from scratch with a meta-objective (Section 4) without rehearsal for fast online continual learning.

**Representation of the Gradient Space in Neural Network**: Since SGD updates lie in the span of input data points (Zhang et al., 2017), gradients (or gradient space) in each layer of the ANN can be represented by low-dimensional basis vectors (Saha et al., 2021b). Let, $\theta_0 \in \mathbb{R}^{C_o \times C_i \times k \times k}$ represent filters in a convolutional (Conv) layer, where $C_i$ ($C_o$) is the number of input (output) channels of that layer and $k$ is the kernel size of the filters. Following Saha et al. (2021b), $\theta_0$ (hence gradient, $\nabla_{\theta_0}\ell$) can be reshaped into a $(C_i \times k \times k) \times C_o$ dimensional matrix. Thus gradients in a Conv layer can be described by $(C_i \times k \times k)$ dimensional space (instead of $C_o \times C_i \times k \times k$). Similarly, if $\theta_0 \in \mathbb{R}^{m \times n}$ represents a weight matrix in a fully-connected (FC) layer where $m$ ($n$) is the dimension of outgoing (incoming) hidden units, the gradient space will be $n$ dimensional (instead of $m \times n$) in this layer.

## 4 CONTINUAL LEARNING WITH AMPHIBIAN

Here, we describe the steps (illustrated in Figure 1) for online continual learning in Amphibian.

**Learning Overview:** At any time $j$ over the learning sequence, Amphibian receives a batch of data, $\mathcal{B}_i \sim \mathcal{D}^{\tau_i}$ with $N_i$ input-label data pairs from current task, $\tau_i$. We aim to update the current model,

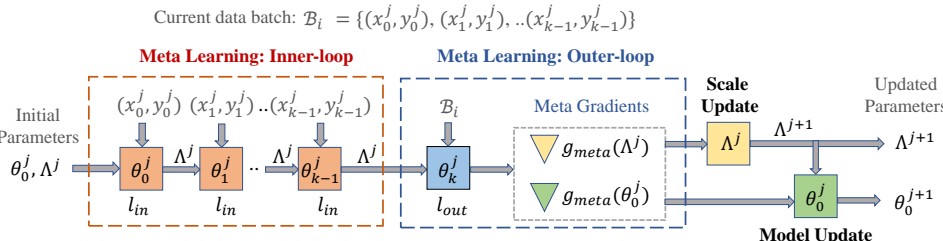

Figure 1: Illustration of continual learning in Amphibian on a batch of data. Given the model, $\theta_0^j$ and scale matrix, $\Lambda^j$, **first** we perform $k$ inner-loop gradient update on $\theta_0^j$ to obtain $\theta_k^j$ with the samples of the current batch, $\beta_i$. In **second** step, we compute the meta-loss on $\theta_k^j$ with the entire batch, $\beta_i$ to obtain meta gradients. In **third** step we update $\Lambda^j$ and **finally** the model, $\theta_0^j$ with these gradients.

$\theta_0^j$ using the update rule:

$$\theta_0^{j+1} = \theta_0^j - \Lambda^{j+1} g_{meta}(\theta_0^j). \tag{4}$$

Here $g_{meta}$ is the gradient obtained from the meta (inner-outer loop) learning process using only the current data, $\mathcal{B}_i$. In our method, we use the gradient space formulation as described in Section 3 and consider the standard bases $(e_i)$ of appropriate dimensions as the bases of gradient space to represent the gradients inside the neural network. We introduce a **scaling matrix**, $\Lambda$ in the update rule. This is a diagonal matrix where each diagonal element, $\lambda_i$ is initialized with $\lambda_i^o$ and then meta-learned simultaneously with $g_{meta}$. Over the continual learning trajectory, $\Lambda$ accumulates the history of gradient alignment among observed data samples. Then it scales the meta-gradient (Equation 4) accordingly to update the model along the direction of positive gradient alignments. Thus $\Lambda$ essentially *learns the learning rate* of the bases of the gradient space during continual learning.

**Meta-Learning Step-1:** At first, on the given batch, $\mathcal{B}_i$ we perform $k$ **inner-loop** updates on $\theta_0^j$ to obtain $\theta_k^j$ as:

$$U_k(\theta_0^j; \mathcal{B}_i) = \theta_k^j = \theta_0^j - \sum_{k'=0}^{k-1} \Lambda^j \nabla_{\theta_{k'}^j} \ell_{in}(\theta_{k'}^j; \mathcal{B}_i[k']). \tag{5}$$

These inner-loop steps differ in two ways from MAML inner-update step (Equation 2). First, for each inner update, we use one sample (if $k = N_i$) or a subset of samples (if $k < N_i$) from $\mathcal{B}_i$ without replacement, whereas MAML uses entire batch ($\mathcal{D}_{in}^i = \mathcal{B}_i$). Second, in our method, each inner gradient is scaled by the $\Lambda^j$ matrix, whereas MAML uses a constant scalar learning rate, $\alpha$. Though meta-learnable per parametric learning rate (Li et al., 2017; Gupta et al., 2020) vector, $\alpha$ and block diagonal preconditioners (Park & Oliva, 2019) have been used in such updates, we learn diagonal $\Lambda$ which differs in dimensions and interpretation.

**Meta-Learning Step-2:** In this online learning setup, unlike MAML, Amphibian does not have access to the data from all the tasks seen so far. Moreover, as Amphibian is a rehearsal-free learner, we can not store past examples in memory and use them to approximately minimize the outer-loop MAML objective (Equation 3) as in Gupta et al. (2020). Instead, in the **outer loop** of meta-learning, we compute the meta-loss, $\ell_{out}$ on current data ($B_i = \mathcal{D}_{out}$) at $\theta_k^j$ and minimize the following objective :

$$\min_{\theta_0^j, \Lambda^j} \mathbb{E}_{\mathcal{B}_i}[\ell_{out}(\theta_k^j; \mathcal{B}_i)] = \min_{\theta_0^j, \Lambda^j} \mathbb{E}_{\mathcal{B}_i}[\ell_{out}(U_k(\theta_0^j, \Lambda^j; \mathcal{B}_i); \mathcal{B}_i)]. \tag{6}$$

Minimizing this objective with respect to $\theta_0^j$ is equivalent to (see Appendix A for full derivation):

$$\min_{\theta_0^j} \mathbb{E}_{\mathcal{B}_i}[\ell_{out}(\theta_k^j; \mathcal{B}_i)] = \min_{\theta_0^j} \left( \ell_{out}(\theta_0^j) - \sum_{i=1}^{M} \lambda_i^j \frac{\partial \ell_{out}(\theta_0^j)}{\partial \theta_0^j} \cdot e_i e_i^T \frac{\partial \ell_{in}(\theta_0^j)}{\partial \theta_0^j} \right), \tag{7}$$

where $M$ is the dimension of gradient space. Here, $\ell_{out}$ is computed on entire batch $\mathcal{B}_i$ while $\ell_{in}$ is computed on a sample (or subset) from $\mathcal{B}_i$. First term on the right-hand side of the objective in Equation 7 minimizes the loss on current batch of data, $\mathcal{B}_i$. The second term encourages positive alignment (inner product) of gradients computed on samples of $\mathcal{B}_i$ along selective gradient basis directions ($e_i$) depending on scale $\lambda_i$. For instance, for positive $\lambda_i$, inner product of data gradients along $e_i$ is maximized whereas for zero or negative $\lambda_i$, such gradient alignments are not encouraged. Gradient of this objective is given by $g_{meta}(\theta_0^j)$ which is used for model update in Equation 4.

**Scale Update Step:** Next, gradient of the meta-objective (in Equation 6) with respect to each scale, $\lambda_i^j$ in $\Lambda^j$ can be simply expressed (using first-order approximation (Finn et al., 2017)) as follows:

$$g_{meta}(\lambda_i^j) = -\frac{\partial \ell_{out}(\theta_k^j)}{\partial \theta_k^j} \cdot e_i e_i^T \sum_{k'=0}^{k-1} \frac{\partial \ell_{in}(\theta_{k'}^j)}{\partial \theta_{k'}^j} = -\ell'_{out}(\theta_k^j) \cdot e_i e_i^T \sum_{k'=0}^{k-1} \ell'_{in}(\theta_{k'}^j). \quad (8)$$

Full derivation is given in Appendix B. Equation 8 denotes that if outer-loop gradient $g_{out} = \ell'_{out}(\theta_k^j)$ and accumulated inner-loop gradient $\bar{g}_{in} = \sum_{k'=0}^{k-1} \ell'_{in}(\theta_{k'}^j)$ has positive inner product (aligned) along $e_i$, then $g_{meta}(\lambda_i^j)$ will be negative, whereas if the inner product is zero (negative) then $g_{meta}(\lambda_i^j)$ will be zero (positive). The scale update rule: $\lambda_i^{j+1} = \lambda_i^j - \eta g_{meta}(\lambda_i^j)$ can be expressed as:

$$\lambda_m^{j+1} = \lambda_m^j + \eta \ell'_{out}(\theta_k^j) \cdot e_m e_m^T \sum_{k'=0}^{k-1} \ell'_{in}(\theta_{k'}^j) = \lambda_m^0 + \eta \sum_j \left( \ell'_{out}(\theta_k^j) \cdot e_m e_m^T \sum_{k'=0}^{k-1} \ell'_{in}(\theta_{k'}^j) \right), \quad (9)$$

where $\eta$ is the learning rate for the scales. This update rule provides two valuable insights. First, the value of a scale will increase (decrease) if, along the corresponding basis, $e_i$ direction outer- and inner-loop gradient trajectories have positive (negative) inner product or alignment (interference). Second, over the entire continual learning sequence up to time $j$, $\lambda_i^j$ accumulates the history of such gradient alignments or interferences. Now if we do not use accumulation and update the scales as $\lambda_i^{j+1} = \lambda_i^0 - \eta g_{meta}(\lambda_i^j)$ with current $g_{meta}(\lambda_i^j)$ and use these scales for model update (Equation 4), we would get fast learning on the current data but model will forget past data. In contrast, by using cumulatively updated (Equation 9) $\Lambda$ in Equation 4, we ensure that the gradient step on each incoming data is accelerated along the direction of positive *cumulative* alignment, whereas blocked along the direction of zero or negative alignments thus minimizing catastrophic forgetting. Therefore, this online accumulation of gradient alignments in scale matrices over the learning sequence enables Amphibian to learn continually with minimum forgetting without any data rehearsal.

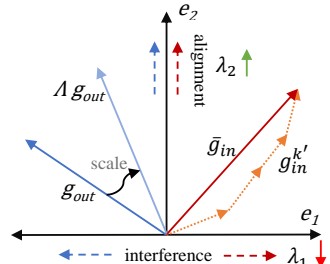

Figure 2: Conceptual illustration. Along $e_2$ ($e_1$) $g_{out}$ and $\bar{g}_{in}$ has alignment (interference), hence $\lambda_2$ ($\lambda_1$) will increase (decrease). Considering, $g_{meta}(\theta_0^j) \approx g_{out}$, $g_{meta}$ is scaled accordingly in model update to reflect alignment history.

**Model Update Step:** Finally, with $g_{meta}(\theta_0^j)$ and the updated scale $\Lambda^{j+1}$ we perform model update on the current batch as in Equation 4. As the scales, $\lambda_i$ can take both positive and negative values, to prevent gradient ascent (both in Equation 4 and inner loop Equation 5) we only use their positive part using $(\lambda_i)_+ = \mathbb{1}_{\lambda_i \geq 0} \lambda_i$ function, where $\mathbb{1}_{\cdot \geq 0} : \mathbb{R} \to \{0, 1\}$. In the meta-learning steps, we ensure differentiability of this function using the straight-through estimator (Bengio et al., 2013; Von Oswald et al., 2021). With the updated model $\theta_0^{j+1}$ and scale matrix $\Lambda^{j+1}$, we (continually) learn the next batch of data. The pseudocode of the algorithm is provided in Algorithm 1 in Appendix C.

## 5 EXPERIMENTAL SETUP

**Datasets and Models.** We evaluate Amphibian and the baselines in online continual learning (OCL) (Mai et al., 2022) setups where models learn from the single pass over the data stream. We use 5 standard image classification benchmarks in continual learning: 20 tasks split **CIFAR-100** (Gupta et al., 2020), 40 tasks split **TinyImagenet** (Deng et al., 2021), 25 tasks split **5-Datasets** (Saha et al., 2021b), 20 tasks split **ImageNet-100** (Yan et al., 2021) and 10 tasks split **miniImageNet** (Shim et al., 2021). Similar to Gupta et al. (2020); Deng et al. (2021), we use 5-layer network for CIFAR-100 and 5-Datasets, and 6-layer network for TinyImagenet. For ImageNet-100 and miniImageNet experiments we use ResNet-18 (Lopez-Paz & Ranzato, 2017) model. Details on the dataset statistics/splits, and network architectures are provided in the Appendix D.1 and D.2 respectively.

**Baselines and Training.** We compare Amphibian with rehearsal-free methods: EWC (Kirkpatrick et al., 2017) and SI (Zenke et al., 2017) which use parametric regularization; GPM (Saha et al., 2021b) and SGP (Saha & Roy, 2023) which use gradient projection; and NCL (Kao et al., 2021) which uses both gradient projection and regularization. Although comparisons with rehearsal-free and rehearsal-based methods are not always fair (especially with large data memory), we compare

Table 1: Performance (mean $\pm$ std from 3 runs with random seeds) comparisons in online continual learning. (*) indicates results for CIFAR-100 and TinyImagenet are taken from Gupta et al. (2020).

| Rehearsal | Methods | CIFAR-100 | | TinyImagenet | | 5-Datasets | |
|---|---|---|---|---|---|---|---|
| | | ACC (%) | BWT (%) | ACC (%) | BWT (%) | ACC (%) | BWT (%) |
| ✓ | ER* | $47.8 \pm 0.73$ | - $12.4 \pm 0.83$ | $39.3 \pm 0.38$ | - $14.3 \pm 0.89$ | $83.4 \pm 0.69$ | - $9.43 \pm 1.24$ |
| | GEM* | $48.2 \pm 1.10$ | - $13.7 \pm 0.70$ | $40.5 \pm 0.79$ | - $13.5 \pm 0.65$ | $86.9 \pm 1.09$ | - $7.43 \pm 0.61$ |
| | A-GEM* | $46.9 \pm 0.31$ | - $13.4 \pm 1.44$ | $38.9 \pm 0.47$ | - $13.6 \pm 1.73$ | $82.4 \pm 1.18$ | - $9.66 \pm 0.82$ |
| | MER* | $51.3 \pm 1.05$ | - $13.4 \pm 1.44$ | $38.9 \pm 0.47$ | - $13.6 \pm 1.73$ | $88.5 \pm 0.28$ | - $7.00 \pm 0.33$ |
| | DER++ | $53.4 \pm 1.75$ | - $8.16 \pm 1.12$ | $44.1 \pm 0.29$ | - $14.5 \pm 0.49$ | $86.7 \pm 0.86$ | - $4.5 \pm 1.09$ |
| | CLS-ER | $60.5 \pm 0.45$ | $0.90 \pm 0.50$ | $51.8 \pm 1.01$ | $15.9 \pm 0.71$ | $89.1 \pm 1.29$ | $7.13 \pm 0.09$ |
| | La-MAML* | $61.1 \pm 1.44$ | - $9.00 \pm 0.20$ | $52.5 \pm 1.35$ | - $3.70 \pm 1.22$ | $89.0 \pm 1.45$ | - $5.97 \pm 2.00$ |
| ✗ | EWC | $50.4 \pm 1.88$ | - $1.53 \pm 0.98$ | $43.6 \pm 0.83$ | - $0.83 \pm 1.17$ | $80.1 \pm 1.76$ | - $8.06 \pm 1.22$ |
| | SI | $51.1 \pm 0.69$ | - $0.73 \pm 0.24$ | $42.3 \pm 3.43$ | - $1.60 \pm 0.14$ | $80.9 \pm 1.71$ | - $5.96 \pm 2.45$ |
| | GPM | $59.1 \pm 1.26$ | - $0.00 \pm 0.12$ | $48.3 \pm 2.69$ | - $0.30 \pm 1.00$ | $84.3 \pm 2.57$ | - $2.06 \pm 0.94$ |
| | SGP | $61.3 \pm 1.25$ | - $0.00 \pm 0.13$ | $52.7 \pm 0.26$ | - $0.00 \pm 0.44$ | $86.1 \pm 3.52$ | - $5.56 \pm 3.20$ |
| | NCL | $56.5 \pm 1.08$ | - $0.00 \pm 0.18$ | $49.7 \pm 0.91$ | - $0.83 \pm 0.05$ | $85.5 \pm 0.58$ | - $0.00 \pm 0.05$ |
| ✗ | **Amphibian** | $\mathbf{65.0 \pm 0.96}$ | - $1.30 \pm 0.25$ | $\mathbf{54.8 \pm 0.60}$ | - $0.72 \pm 0.22$ | $\mathbf{89.3 \pm 1.24}$ | - $4.87 \pm 0.96$ |

with : ER (Chaudhry et al., 2019b), GEM (Lopez-Paz & Ranzato, 2017), A-GEM (Chaudhry et al., 2019a), DER++ (Buzzega et al., 2020) and CLS-ER (Arani et al., 2022) having moderate data memory (100 to 400 samples). We also compare with MER (Riemer et al., 2019) and La-MAML (Gupta et al., 2020) which uses meta-learning and memory rehearsal for OCL. Following baselines, we do not use any offline pre-trained models, rather we train models from scratch. In Amphibian, scale learning rate ($\eta$) and initial scale value ($\lambda_i^0$) hyperparameters were set with grid search (as in Gupta et al. (2020)) with held out validation data from training sets. Similarly, all the hyperparameters of the baselines were tuned. Details of training setup, implementations and a list of all the hyperparameters considered in the baselines and our method is given in Appendix D.

**Evaluation Metrics.** We measure OCL performance with two metrics: **ACC** - measures average test classification accuracy of all tasks and **BWT** (backward transfer) - measures influence of new learning on the past knowledge with negative BWT denotes forgetting. They are defined as:

$$\text{ACC} = \frac{1}{T} \sum_{i=1}^{T} R_{T,i}; \quad \text{BWT} = \frac{1}{T-1} \sum_{i=1}^{T-1} R_{T,i} - R_{i,i}. \tag{10}$$

Here, $T$ is the total number of tasks and $R_{T,i}$ is the accuracy of the model on $i^{th}$ task after learning the $T^{th}$ task. Higher ACC in online setup signifies fast continual learning ability, however to gain better insight on the fast learning ability we introduce two additional metrics - task learning efficiency (**TLE**) and few-shot forward transfer (**FWT**$_{FS}$). TLE (Finn et al., 2019) is defined as the size of $\mathcal{D}_t$ required to achieve certain ($\gamma\%$) classification accuracy on that task, $t$. This metric implies if less data (small TLE) is required to achieve a certain performance level, then the model can transfer knowledge faster, hence is a fast learner. For comparisons among the baselines, we set $\gamma$ as 90% of the final Amphibian accuracies on each task, $t$. Additionally, in meta-learning (Finn et al., 2017; Nichol et al., 2018), fast learning capability is measured with *N-way K-shot* adaptation accuracy where a meta-learned model is trained on $NK$ examples with few gradient steps and then tested. Thus a truly fast learner should also perform well under this setup. To evaluate this, after learning each OCL task, $t$ we sample $K$ examples from each of the $N$ classes from the next task and fully adapt the model for $n_s$ steps. Then record the test accuracy on this new task as $FWT_{FS}^t$ - the few-shot forward transfer capacity of the model learned after task $t$. For all the tasks we measure and compare this capacity as $\mathbf{FWT}_{FS} = \frac{1}{T} \sum_{t=0}^{T-1} FWT_{FS}^t$.

## 6 RESULTS AND ANALYSES

### 6.1 CONTINUAL LEARNING AND FAST LEARNING PERFORMANCE COMPARISONS

**Accuracy and Forgetting. First**, we evaluate and compare the performance in task-incremental OCL setups where each task has a separate classifier and task identity is used during inference. In Table 1, we provide comparisons of ACC and BWT among various methods within this setup across 3 different datasets. Among the rehearsal-based methods, meta-learner La-MAML achieves better accuracy. Even without memory rehearsal, Amphibian outperforms La-MAML (by up to ~8.5%

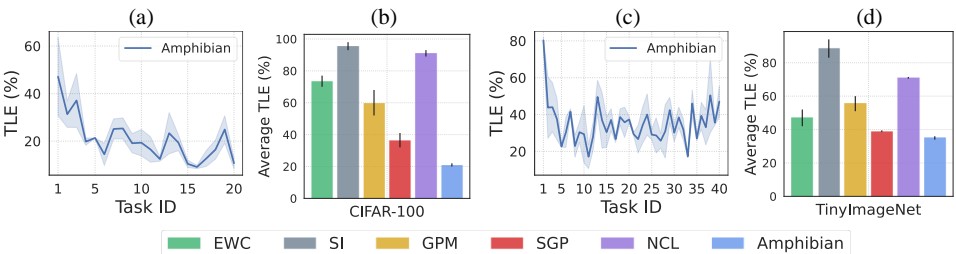

Figure 3: (a) Task learning efficiency (TLE) of Amphibian. (b) Average TLE of all the methods for CIFAR-100 tasks. (c) TLE of Amphibian and (d) Average TLE of all the methods for TinyImagenet.

in ACC) with less forgetting. Since the rehearsal-free methods are under-explored for OCL and Amphibian is a rehearsal-free method, in the following analyses and discussions we focus primarily on rehearsal-free methods. EWC and SI achieve similar performance, however, compared to Amphibian they severely underperform (by up to ∼14% in ACC). This shows that the parametric importance-based regularization used in these methods to ensure stability of knowledge (as indicated by small forgetting in Table 1) is not favorable for fast learning required in OCL. In contrast, gradient projection methods such as GPM and SGP achieve up to ∼10% ACC gain over EWC and SI. However, Amphibian outperforms these methods by up to ∼4% in ACC with marginally higher forgetting. This shows without any explicit constraints of orthogonal gradient projections (as in GPM and SGP), Amphibian properly *learned to scale* the gradient updates from the online data to perform better in OCL. NCL performance lies in between gradient projection and regularization-based methods and is outperformed by Amphibian. Moreover, in challenging 5-Dataset tasks where data arrives from dissimilar domains over time, Amphibian obtains the best performance over all the methods. **Next**, in Table 2, we compare Amphibian's performance with the baselines on 20 split ImageNet-100 tasks using the ResNet-18 model. Here, Amphibian archives ∼ 2.4% better accuracy, demonstrating its scalability to larger datasets and complex networks. **Finally**, we evaluate our method in a class-incremental OCL setup. Following Shim et al. (2021), we train a ResNet-18 model from scratch on 10 miniImageNet tasks and during inference, task identity is not used. In Figure 4(a), we compare the average accuracy of the model trained with different rehearsal-based and free methods. After 10 tasks, our method achieves better accuracy than the best rehearsal-based method - CLS-ER. All the following analyses are performed in task-incremental setups.

**Task Learning Efficiency (TLE).** As discussed in Section 5, a fast online continual learner should achieve high performance with less amount of data (smaller TLE per task). In Figure 3(a) we show TLE of Amphibian for each CIFAR-100 task. Averaging over all the tasks we obtain an average TLE of 21% for Amphibian. This means, on average, a new task in Amphibian can be learned to 90% of its final achievable accuracy by learning on 21% of the data from that task. In Figure 3(c) TLE of Amphibian for each TinyImageNet task is shown which gives an Average

Table 2: Performance comparisons for 20 IamgeNet-100 tasks on ResNet-18.

| | ImageNet-100 | |
|---|---|---|
| **Methods** | ACC (%) | BWT(%) |
| GPM | 45.5 ± 1.33 | - 0.00 ± 0.23 |
| La-MAML | 51.7 ± 1.15 | - 5.60 ± 0.80 |
| **Amphibian** | **54.1 ± 0.76** | **- 0.34 ± 0.20** |

TLE of 35%. We compare the Average TLE of all the rehearsal-free methods in Figure 3(b) and (d). In both datasets, Amphibian outperforms all the other methods, indicating that Amphibian can obtain a high accuracy level on given tasks very quickly by observing fewer examples from the data stream, hence it is a fast learner.

**Few-shot Forward Transfer (FWT$_{FS}$).** In Figure 4(b) we compare FWT$_{FS}$ of different algorithms for CIFAR-100 and TinyImagenet datasets, where higher FWT$_{FS}$ means better few-shot (rapid) learner. For each FWT$_{FS}^t$, we use *5-way 5-shot* training data and adapt the network with $n_s = 10$ steps. Compared to the rehearsal-free methods, Amphibian achieves up to ∼ 5% better FWT$_{FS}$, which demonstrates the fast learning ability developed in the model from online continual learning in Amphibian. Here we show the performance of (meta-learned) La-MAML, which also achieves better performance than other rehearsal-free baselines but outperformed by Amphibian. This shows meta-learning plays a key role in building fast learning capability in Amphibian and La-MAML.

**Memory overhead and Training Times**. In Figure 4(c) we compare the memory overhead during training in each method for either storing the old model, important parameters and/or past data. The numbers are normalized by the memory overhead of Amphibian. For both datasets, all the baseline

Figure 4: (a) Average accuracy for miniImageNet tasks in online class-incremental learning.(b) Few-shot forward transfer, $\text{FWT}_{FS}$ (higher the better). (c) Training time memory overhead comparisons.

Table 3: Comparison of Amphibian with candidate rehearsal-free meta-learning methods for OCL.

| | **CIFAR-100** | | | **TinyImagenet** | | |
|---|---|---|---|---|---|---|
| **Methods** | ACC (%) | BWT(%) | $\text{FWT}_{FS}$(%) | ACC (%) | BWT(%) | $\text{FWT}_{FS}$(%) |
| **Amphibian** | $65.0 \pm 0.96$ | $-1.30 \pm 0.25$ | $47.6 \pm 1.97$ | $54.8 \pm 0.60$ | $-0.72 \pm 0.22$ | $40.6 \pm 0.87$ |
| La-MAML (No Rehearsal) | $56.1 \pm 0.87$ | $-12.8 \pm 0.17$ | $46.7 \pm 1.89$ | $49.2 \pm 0.55$ | $-7.23 \pm 0.16$ | $39.5 \pm 0.75$ |
| Online Meta-SGD | $51.3 \pm 2.54$ | $-15.7 \pm 2.50$ | $45.9 \pm 1.44$ | $45.1 \pm 1.31$ | $-10.7 \pm 0.57$ | $37.7 \pm 0.44$ |
| Online MAML | $49.0 \pm 1.81$ | $-18.0 \pm 2.00$ | $42.2 \pm 1.48$ | $40.4 \pm 0.57$ | $-8.10 \pm 0.74$ | $31.8 \pm 0.45$ |

methods use orders of magnitude more memory than Amphibian. For instance, where Amphibian only requires up to $\sim 0.5\%$ extra memory compared to network size for gradient scale ($\lambda$) storage (scale numbers given in Appendix E.3), La-MAML requires $100\%$ extra memory for per parametric learning rate storage and up to $179\%$ extra memory for replay buffer. Among all the methods NCL has the highest memory overhead for model and projection matrix storage. Thus Amphibian most memory-efficient learner for OCL. Training time comparisons are provided in Appendix E.1.

## 6.2 ANALYSIS OF AMPHIBIAN: ABLATION STUDIES, AMPHIBIAN-$\beta$, LOSS LANDSCAPES

**Amphibian vs. Online Rehearsal-free Meta-Learning.** To our knowledge, there are no rehearsal-free meta-learners for OCL, so we adapt popular meta-learning approaches for OCL and compare them with Amphibian in Table 3. First method is **La-MAML(No Rehearsal)**, where we remove the memory buffer from La-MAML and compute the meta-loss with only current data (as Amphibian). The notable difference between this method and the Amphibian is that it learns *learning rates for each parameter* and uses that in inner- and outer-loop gradient updates, whereas Amphibian learns *a diagonal scale matrix at each layer* and scale the gradient directions accordingly. From Table 3, we find that La-MAML(No Rehearsal) vastly under-performs Amphibian, particularly it suffers from large forgetting (up to $\sim 11.5\%$ more than Amphibian). This shows the novel meta-objective optimization and gradient scaling with the learned scale matrix ($\Lambda$) enable Amphibian to learn continually without rehearsal. In second method, we replace the meta-learned learning rate in outer-loop update from La-MAML(No Rehearsal) with a constant learning rate, whereas inner-loop still uses learnable learning rates. This converts the method to **online Meta-SGD** (Li et al., 2017). Results show that ACC drops further, forgetting increases and fast learning capability reduces. Finally, we use constant inner- and outer-loop learning rates in online Meta-SGD, which converts the method to **online MAML** (Finn et al., 2017). With no learnable scale matrix or learning rate to encode the history of past data, this method performs the worst in OCL setup. In summary, these analyses clearly show the functional difference between our method and La-MAML and MAML, and highlight the importance of meta-learned gradient scaling in Amphibian for fast OCL performance.

**Amphibian-$\beta$ .** Here, we introduce a regularized version of Amphibian - **Amphibian-$\beta$** to understand the relationship between fast learning and forgetting. Amphibian-$\beta$ uses the same online continual learning steps as Amphibian except it performs model update using the following rule:

$$\theta_0^{j+1} = \theta_0^j - \Lambda^{j+1} g_{meta}(\theta_0^j) - \beta(\theta_0^j - \theta_0^0) = \theta_0^0 - \sum_{j'=0}^{j} (1-\beta)^{j-j'} \Lambda^{j'+1} g_{meta}(\theta_0^{j'}). \quad (11)$$

So far, used $j$ was used as the time index over the entire learning sequence (spreading across tasks). Here, we denote $j$ as $j^{th}$ time step at task $\tau_i$. Thus, at the start of task $\tau_i$, $\theta_0^0$ would denote optimum parameter learned till task $\tau_{i-1}$. Here, $\beta \in [0, 1]$. For $\beta = 0$ it reduces to Amphibian update whereas for $\beta = 1$ the model parameter stays at initial point $\theta_0^0$. Thus increasing $\beta$ from 0 we can regularize the Amphibian update to stay near the solution of the past tasks. This is a useful concept in continual

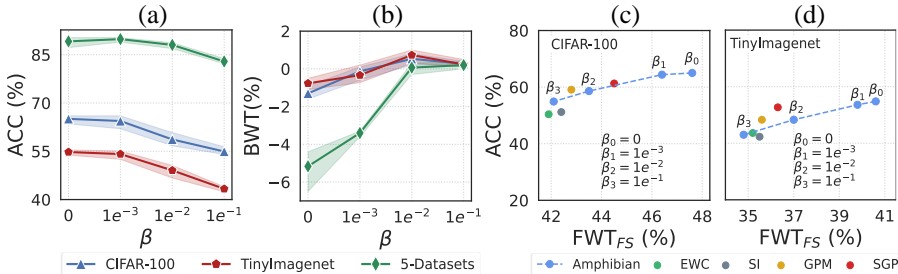

Figure 5: Variations of (a) ACC, (b) BWT, with $\beta$ in Amphibian-$\beta$. ACC vs. $\text{FWT}_{FS}$ comparisons for various Amphibian-$\beta$ with rehearsal-free baselines for (c) CIFAR-100 and (d) TinyImagenet.

learning as staying close to the old solution point provides a degree of protection against catastrophic forgetting. In Figure 5(a) and (b) we show how ACC and BWT vary when $\beta$ is varied (from 0 to 0.1). For all datasets, increasing $\beta$ reduces forgetting in Amphibian (Figure 5(b)), especially for $\beta > 1e^{-2}$ there is no forgetting. However, with increasing $\beta$, ACC degrades (Figure 5(a)). These results indicate that explicit regularization for forgetting reduction restricts the fast continual learning capacity of Amphibian. In Figure 5(c) and (d) we plot ACC vs $\text{FWT}_{FS}$ for Amphibian-$\beta$ with rehearsal-free baselines for CIFAR-100 and TinyImagenet. These plots show as we increase $\beta$, both continual and fast learning performance of Amphibian becomes similar to the baselines. Such functional similarities provide valuable insight that explicit regularization or constraints used in these rehearsal-free methods primarily focus on forgetting mitigation at the expense of fast learning, hence they underperform in OCL setup. These analyses call for a rethinking of regularization or constraint design in rehearsal-free methods and provide motivation for exploring Amphibian-like learner that learns the required constraints for optimal balance between fast and continual learning.

**Understating Continual Learning Dynamics in Amphibian.** Continual learning works on the principle that learning a new task should not (or minimally) increase the loss of the old tasks. Amphibian does not explicitly minimize any such objectives, instead, it continually updates models in the direction of positive cumulative gradient alignments among the observed data. To understand how Amphibian enables continual learning we use loss landscape visualizations (Verwimp et al., 2021; Mirzadeh et al., 2020). In Figure 6(a) we plot the loss contour of task 1 from CIFAR-100 in 2D plane. Here $\theta^t$ indicates the network model after learning task $t$. Here, the black line indicates the learning trajectory, with each cross

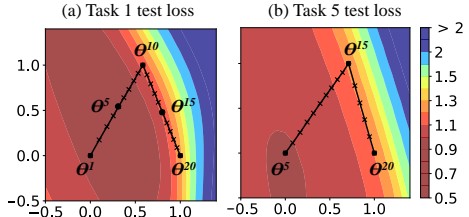

Figure 6: Dynamics of continual learning in Amphibian. Loss contour of (a) Task 1 and (b) Task 5 plotted on 2D planes defined by parameters $(\theta^1, \theta^{10}, \theta^{20})$ and $(\theta^5, \theta^{15}, \theta^{20})$ respectively.

point representing the projection of learned models on the plane along the trajectory. In Amphibian when we sequentially learn from task 1 to task 20, along the learning trajectory ($\theta^1 \rightarrow \theta^{10} \rightarrow \theta^{20}$ in Figure 6(a)) loss of task 1 only increase minimally from the initial point ($\theta^1$). A similar trend can be seen for task 5 (in Figure 6(b), where along the learning trajectory ($\theta^5 \rightarrow \theta^{15} \rightarrow \theta^{20}$) loss increases in task 5 is minimal. Similar pattern is also found for TinyImagenet tasks (Appendix E.2). Thus, in Amphibian, model updates along the directions with positive cumulative gradient alignments prevent a significant increase in the loss of past data enabling continual learning with minimum forgetting.

## 7 CONCLUSIONS

In this paper, we introduce a rehearsal-free meta-learner - Amphibian that in a fully online manner learns to scale the gradient updates to enable fast online continual learning. To this end, Amphibian optimizes a novel meta-objective and learns scale matrices that accumulate the history of gradient alignments among the data samples observed over the learning trajectory. Using these scale matrices it updates the model in the direction of positive cumulative gradient alignments. On various continual image classification tasks, we show that such meta-learned scaled gradient update in Amphibian enables memory-efficient, data-efficient, and fast online continual learning. In conclusion, we believe Amphibian offers a unified framework for exploring meta-learning and continual learning, making it a valuable tool for dissecting inherent trade-offs and ultimately facilitating the development of improved algorithms that strike a desired balance between fast and continual learning.

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

APPENDIX

Derivation of Amphibian meta-objective and meta-gradients are provided in Section A and B respectively. Pseudocode of the Amphibian algorithm is given in Section C. Experimental details including dataset statistics, network architectures, a list of hyperparameters along with implementation details are provided in Section D. Additional results and analyses are provided in Section E. **Amphibian source codes are attached as supplementary material in the 'Amphibian_CODE' folder**.

## A    DERIVATION OF AMPHIBIAN META-OBJECTIVE

In this section, we will show that when we optimize the following meta-objective in Amphibian:

$$\min_{\theta_0^j} \mathbb{E}_{\mathcal{B}_i}[\ell_{out}(\theta_k^j; \mathcal{B}_i)] = \min_{\theta_0^j} \mathbb{E}_{\mathcal{B}_i}[\ell_{out}(U_k(\theta_0^j, \Lambda^j; \mathcal{B}_i); \mathcal{B}_i)]. \tag{12}$$

where each of the $k$ inner-update is taken using a sample (or subset of samples) from current batch, $\mathcal{B}_i$ from task $\tau_i$ and the meta-loss, $\ell_{out}$ is computed on the entire current batch data,$\mathcal{B}_i$, it is equivalent to minimizing the following objective:

$$\min_{\theta_0^j} \left( \ell_{out}(\theta_0^j) - \sum_{m=1}^{M} \lambda_m^j \frac{\partial \ell_{out}(\theta_0^j)}{\partial \theta_0^j} \cdot e_m e_m^T \frac{\partial \ell_{in}(\theta_0^j)}{\partial \theta_0^j} \right). \tag{13}$$

For that, let us define,

$$g_k = \frac{\partial \ell_{out}(\theta_k^j)}{\partial \theta_k^j} \quad \text{(gradient of meta-loss at } \theta_k^j) \tag{14}$$

$$\bar{g}_k = \frac{\partial \ell_{out}(\theta_0^j)}{\partial \theta_0^j} \quad \text{(gradient of meta-loss at } \theta_0^j) \tag{15}$$

$$g_{k'} = \frac{\partial \ell_{in}(\theta_{k'}^j)}{\partial \theta_{k'}^j} \quad \text{(gradient of inner-loss at } \theta_{k'}^j, \text{ where } k' < k) \tag{16}$$

$$\bar{g}_{k'} = \frac{\partial \ell_{in}(\theta_0^j)}{\partial \theta_0^j} \quad \text{(gradient of inner-loss at } \theta_0^j, \text{ where } k' < k) \tag{17}$$

$$\theta_{k'+1}^j = \theta_{k'}^j - \Lambda^j g_{k'} \quad \text{(sequence of parameter vectors)} \tag{18}$$

$$\bar{H}_k = \ell_{out}''(\theta_0^j) \quad \text{(Hessian of meta-loss at } \theta_0^j) \tag{19}$$

$$\bar{H}_{k'} = \ell_{in}''(\theta_0^j) \quad \text{(Hessian of inner-loss at } \theta_0^j) \tag{20}$$

First, let's write the gradient of meta-loss (outer-loop loss) at $\theta_k^j$ from Equation 14 as (using Taylor's expansion (Nichol et al., 2018)):

$$g_k = \ell_{out}'(\theta_k^j) = \ell_{out}'(\theta_0^j) + \ell_{out}''(\theta_0^j)(\theta_k^j - \theta_0^j) + O(||\theta_k^j - \theta_0^j||^2) \quad (' \text{ implies derivative w.r.t argument})$$

$$= \bar{g}_k + \bar{H}_k(\theta_k^j - \theta_0^j) + O(||\theta_k^j - \theta_0^j||^2) \quad \text{(using definition of } \bar{g}_k, \bar{H}_k)$$

$$= \bar{g}_k - \bar{H}_k \sum_{k'=0}^{k-1} \Lambda^j g_{k'} + O(\Lambda^2) \quad \text{(using } \theta_k^j - \theta_0^j = -\sum_{k'=0}^{k-1} \Lambda^j g_{k'}, \text{ from Equation 5)}$$

$$= \bar{g}_k - \bar{H}_k \sum_{k'=0}^{k-1} \Lambda^j \bar{g}_{k'} + O(\Lambda^2) \quad \text{(using } g_{k'} = \bar{g}_{k'} + O(\Lambda))$$

$$= \bar{g}_k - \bar{H}_k \sum_{k'=0}^{k-1} \bar{g}_{k'}^{\Lambda} + O(\Lambda^2) \quad \text{(let scaled update, } \bar{g}_{k'}^{\Lambda} = \Lambda^j \bar{g}_{k'})$$

$$\tag{21}$$

Now, let's derive the meta-gradient (or MAML gradient (Finn et al., 2017)) for parameters $\theta_0^j$, denoted as $g_{meta}(\theta_0^j)$:

$$
\begin{aligned}
g_{meta}(\theta_0^j) &= \frac{\partial \ell_{out}(\theta_k^j)}{\partial \theta_0^j} = \frac{\partial \ell_{out}(\theta_k^j)}{\partial \theta_k^j} \frac{\partial U(\theta_{k-1}^j)}{\partial \theta_0^j} \\
&= g_k \frac{\partial U(\theta_{k-1}^j)}{\partial \theta_{k-1}^j} ... \frac{\partial U(\theta_0^j)}{\partial \theta_0^j} \quad \text{(repeatedly applying chain rule and using, } \theta_k^j = U(\theta_{k-1}^j) \text{ )} \\
&= \prod_{k'=0}^{k-1} \left( \frac{\partial}{\partial \theta_{k'}^j} (\theta_{k'}^j - \Lambda^j g_{k'}) \right) g_k \\
&= \prod_{k'=0}^{k-1} \left( \frac{\partial}{\partial \theta_{k'}^j} (\theta_{k'}^j - g_{k'}^\Lambda) \right) g_k \\
&= \prod_{k'=0}^{k-1} \left( I - H_{k'}^\Lambda \right) g_k \quad \text{(where, } H_{k'}^\Lambda \text{ is Hessian of scaled gradient, } g_{k'}^\Lambda \text{ at } \theta_{k'}^j)
\end{aligned}
$$

(22)

Using Taylor's theorem and dropping higher order ($O(\Lambda^2)$) terms (Nichol et al., 2018), we can write $H_{k'}^\Lambda \approx \bar{H}_{k'}^\Lambda$ and then using $g_k$ from Equation 21 in Equation 22 we get:

$$
\begin{aligned}
g_{meta}(\theta_0^j) &= \left( \prod_{k'=0}^{k-1} (I - \bar{H}_{k'}^\Lambda) \right) \left( \bar{g}_k - \bar{H}_k \sum_{k'=0}^{k-1} \bar{g}_{k'}^\Lambda \right) + O(\Lambda^2) \\
&= \left( I - \sum_{k'=0}^{k-1} \bar{H}_{k'}^\Lambda) \right) \left( \bar{g}_k - \bar{H}_k \sum_{k'=0}^{k-1} \bar{g}_{k'}^\Lambda \right) + O(\Lambda^2) \\
&= \bar{g}_k - \sum_{k'=0}^{k-1} \bar{H}_{k'}^\Lambda \bar{g}_k - \bar{H}_k \sum_{k'=0}^{k-1} \bar{g}_{k'}^\Lambda + O(\Lambda^2)
\end{aligned}
$$

(23)

Now, using $k = 1$ in Equation 23 we can derive the equivalent objective in Equation 13. For higher $k$, the form of objective becomes complicated but has a similar set of terms. Thus putting $k = 1$ in Equation 23:

$$
\begin{aligned}
\frac{\partial \ell_{out}(\theta_k^j)}{\theta_0^j} &= g_{meta}(\theta_0^j) = \bar{g}_1 - \bar{H}_0^\Lambda \bar{g}_1 - \bar{H}_1 \bar{g}_0^\Lambda + O(\Lambda^2) \\
&= \bar{g}_1 - \frac{\partial}{\partial \theta_0^j} (\bar{g}_1 \cdot \bar{g}_0^\Lambda) \quad \text{(using } \frac{\partial}{\partial \theta_0^j} (\bar{g}_1 \cdot \bar{g}_0^\Lambda) = \bar{H}_0^\Lambda \bar{g}_1 + \bar{H}_1 \bar{g}_0^\Lambda) \\
&= \frac{\partial \ell_{out}(\theta_0^j)}{\partial \theta_0^j} - \frac{\partial}{\partial \theta_0^j} \left( \frac{\partial \ell_{out}(\theta_0^j)}{\partial \theta_0^j} \cdot \Lambda^j \frac{\partial \ell_{in}(\theta_0^j)}{\partial \theta_0^j} \right) \quad \text{(expressing terms as derivatives)} \\
&= \frac{\partial}{\partial \theta_0^j} \left( \ell_{out}(\theta_0^j) - \sum_{m=1}^{M} \lambda_m^j \frac{\partial \ell_{out}(\theta_0^j)}{\partial \theta_0^j} \cdot e_m e_m^T \frac{\partial \ell_{in}(\theta_0^j)}{\partial \theta_0^j} \right),
\end{aligned}
$$

(24)

which is precisely the gradient of the Amphibian meta-objective in Equation 13.

## B    DERIVATION OF META-GRADIENTS

In this section we first derive the **meta-gradients of scales** $\lambda_m^j$ which is defined as:

$$g_{meta}(\lambda_m^j) = \frac{\partial \ell_{out}(\theta_k^j)}{\partial \lambda_m^j} = \frac{\partial \ell_{out}(\theta_k^j)}{\partial \theta_k^j} \cdot \frac{\partial \theta_k^j}{\partial \lambda_m^j}$$

$$= \frac{\partial \ell_{out}(\theta_k^j)}{\partial \theta_k^j} \cdot \frac{\partial}{\partial \lambda_m^j}\left( U(\theta_{k-1}^j) \right)$$

$$= \frac{\partial \ell_{out}(\theta_k^j)}{\partial \theta_k^j} \cdot \frac{\partial}{\partial \lambda_m^j}\left( \theta_{k-1}^j - (\lambda_m^j)_+ e_m e_m^T \frac{\partial \ell_{in}(\theta_{k-1}^j)}{\partial \theta_{k-1}^j} \right)$$

$$= \frac{\partial \ell_{out}(\theta_k^j)}{\partial \theta_k^j} \cdot \left( \frac{\partial}{\partial \lambda_m^j}\theta_{k-1}^j - \frac{\partial}{\partial \lambda_m^j}\left( (\lambda_m^j)_+ e_m e_m^T \frac{\partial \ell_{in}(\theta_{k-1}^j)}{\partial \theta_{k-1}^j} \right) \right)$$

$$= \frac{\partial \ell_{out}(\theta_k^j)}{\partial \theta_k^j} \cdot \left( \frac{\partial}{\partial \lambda_m^j}\theta_{k-1}^j - e_m e_m^T \frac{\partial \ell_{in}(\theta_{k-1}^j)}{\partial \theta_{k-1}^j}\frac{\partial (\lambda_m^j)_+}{\partial \lambda_m^j} \right)$$

(taking $\dfrac{\partial \ell_{in}(\theta_{k-1}^j)}{\partial \theta_{k-1}^j}$ as constant w.r.t $\lambda_m^j$ to get the first-order MAML approximation

as in Nichol et al. (2018); Gupta et al. (2020))

$$= \frac{\partial \ell_{out}(\theta_k^j)}{\partial \theta_k^j} \cdot \left( \frac{\partial}{\partial \lambda_m^j}\theta_{k-1}^j - e_m e_m^T \frac{\partial \ell_{in}(\theta_{k-1}^j)}{\partial \theta_{k-1}^j} \right)$$

($\dfrac{\partial (\lambda_m^j)_+}{\partial \lambda_m^j}$ is equal to identity using approximations from straight-through estimation

as in Bengio et al. (2013); Von Oswald et al. (2021))

$$= \frac{\partial \ell_{out}(\theta_k^j)}{\partial \theta_k^j} \cdot \left( \frac{\partial}{\partial \lambda_m^j}U(\theta_{k-2}^j) - e_m e_m^T \frac{\partial \ell_{in}(\theta_{k-1}^j)}{\partial \theta_{k-1}^j} \right)$$

$$= \frac{\partial \ell_{out}(\theta_k^j)}{\partial \theta_k^j} \cdot \left( \frac{\partial}{\partial \lambda_m^j}\theta_0^j - e_m e_m^T \sum_{k'=0}^{k-1} \frac{\partial \ell_{in}(\theta_{k'}^j)}{\partial \theta_{k'}^j} \right)$$

(repeatedly expanding and differentiating the update function $U(.)$)

$$g_{meta}(\lambda_m^j) = -\frac{\partial \ell_{out}(\theta_k^j)}{\partial \theta_k^j} \cdot e_m e_m^T \sum_{k'=0}^{k-1} \frac{\partial \ell_{in}(\theta_{k'}^j)}{\partial \theta_{k'}^j}$$

(assuming initial parameters, $\theta_0^j$ at time $j$ is constant w.r.t $\lambda_m^j$)

$$(25)$$

This meta-gradient is used in Equation 9 for scale updates.

Next to obtain the **meta-gradients of (weight) parameters**, $\theta_0^j$ lets recall Equation 22:

$$g_{meta}(\theta_0^j) = \prod_{k'=0}^{k-1}\left( I - H_{k'}^\Lambda \right) g_k = \prod_{k'=0}^{k-1}\left( \frac{\partial}{\partial \theta_{k'}^j}(\theta_{k'}^j - \Lambda^j \frac{\partial \ell_{in}(\theta_{k'}^j)}{\partial \theta_{k'}^j}) \right) g_k \qquad (26)$$

Setting all the first-order terms as constant in the right-hand side of this equation (to ignore the second-order derivatives), we get the first-order approximation of the meta-gradient as:

$$g_{meta}^{FO}(\theta_0^j) = g_k = \ell_{out}'(\theta_k^j), \qquad (27)$$

which is used in model updates in Equation 4. This approximation drastically reduces memory consumption while preserving the performance. This allows scaling of Amphibian to the larger networks with complex datasets (Table 2).

## C  Amphibian Algorithm

The pseudocode of the Amphibian algorithm is provided in Algorithm 1. For the given online batch of data, $\mathcal{B}_i$, Amphibian performs $k$ inner-loop updates, where in each update it uses a sample (or a subset of samples) from $\mathcal{B}_i$ (Line 8). On the final parameters obtained after inner-updates, $\theta_k^j$, it evaluates meta-loss on the entire batch, $\mathcal{B}_i$ to obtain meta-gradients for scales, $\nabla_{\lambda_m^j} \ell_{out}(\theta_k^j; \mathcal{B}_i)$ and weights, $\nabla_{\theta_0^j} \ell_{out}(\theta_k^j; \mathcal{B}_i)$. With this meta-gradients, Ampibian first update the scales (Line 10) and then model weights (Line 11). In inner-updates (Line 8) and outer-update (Line 11), to avoid gradient ascent, only positive parts of the scales are used using: $(\lambda_m)_+ = \mathbb{1}_{\lambda_m \geq 0} \lambda_m$ function, where $\mathbb{1}_{\geq 0} : \mathbb{R} \to \{0, 1\}$. For bias parameters in each layer, we use $\boldsymbol{\lambda}$ vectors instead of diagonal $\Lambda$ matrices in inner-updates (Line 8) and outer-update (line 11). Each element of $\boldsymbol{\lambda}$, learns the learning rate of the corresponding bias parameter which is updated using the similar update step in Line 10.

---

**Algorithm 1** Amphibian Algorithm for Online Continual Learning

---

1: **Inputs**: $\theta$: neural network parameters (weights), $\ell_{in}$: inner-objective, $\ell_{out}$: outer (meta) objective, $\lambda^0$: initial values for all the scales, $\eta$: scale learning rate, $T$: number of tasks.
2: $j \leftarrow 0, \theta_0^0 \leftarrow \theta$          ▷ Initialize
3: $\Lambda^0 \leftarrow \texttt{initialize}\,(\lambda^0)$ ▷ For weights/filters in each layer initialize diagonal matrix $\Lambda^0$ with $\lambda^0$
4: **for** $\tau_i \in 1, 2, \ldots, T$ **do**
5:     **for** batch, $\mathcal{B}_i \sim \mathcal{D}^{\tau_i}$ **do**          ▷ $\mathcal{D}^{\tau_i}$ is data stream of current task $\tau_i$
6:         $k \leftarrow \texttt{size}(\mathcal{B}_i)$
7:         **for** $k' = 0$ **to** $k - 1$ **do**
8:             $\theta_{k'+1}^j = \theta_{k'}^j - \Lambda^j \nabla_{\theta_{k'}^j} \ell_{in}(\theta_{k'}^j; \mathcal{B}_i[k'])$          ▷ Inner-loop updates
9:         **end for**
10:         $\lambda_m^{j+1} = \lambda_m^j - \eta \nabla_{\lambda_m^j} \ell_{out}(\theta_k^j; \mathcal{B}_i)$          ▷ Update scales in $\Lambda^j$
11:         $\theta_0^{j+1} = \theta_0^j - \Lambda^{j+1} \nabla_{\theta_0^j} \ell_{out}(\theta_k^j; \mathcal{B}_i)$          ▷ Update model parameters
12:         $j \leftarrow j + 1$
13:     **end for**
14: **end for**

---

## D  Experimental Details

### D.1  Dataset Splits and Statistics

Split CIFAR-100 has 20 tasks each having 5 distinct classes from CIFAR-100 (Krizhevsky, 2009). Split TinyImagenet has 40 tasks where each task has 5 distinct classes from TinyImagenet-200 (Le & Yang, 2015). Finally, we use a sequence of 5-Datasets including CIFAR-10, MNIST, SVHN, Fashion MNIST, and notMNIST where each dataset is split into five tasks (each having a 2 classes) to obtain a total of 25 tasks in the split 5-Datasets sequence. Dataset statistics used in these experiments are given in Table 4 and 5. Split miniImageNet (Shim et al., 2021) consists of splitting the miniImageNet dataset (Vinyals et al., 2016) into 10 disjoint tasks, where each task contains 10 classes. Here each image is of size $3 \times 84 \times 84$. ImageNet-100 is built by selecting 100 classes from the ImageNet-1k (Deng et al., 2009) dataset. Split ImageNet-100, which is used in our experiment, consists of splitting the ImageNet-100 into 20 disjoint tasks, where each task contains 5 classes. Here each image is of size $3 \times 224 \times 224$. For both split miniImageNet and ImageNet-100, $2\%$ training data from each task is kept aside as validation sets.

### D.2  Network Architecture Details

For split CIFAR-100 experiments, similar to La-MAML (Gupta et al., 2020), we used a 5-layer neural network with 3 convolutional layers each having 160 filters with $3 \times 3$ kernels, followed by two fully connected layers having 320 units each. For split 5-Datasets, we used a 5-layer neural network with 3 convolutional layers each having 200 filters with $3 \times 3$ kernels, followed by two fully connected layers having 400 units each. For split TinyImagenet experiments, similar to La-MAML (Gupta et al., 2020), we used a 6-layer neural network with 4 convolutional layers each

Table 4: Dataset Statistics. 10% training data from each task is kept aside as validation sets.

|  | Split CIFAR-100 | Split Tinyimagenet | Split 5-Datasets |
|---|---|---|---|
| num. of tasks | 20 | 40 | 25 |
| input size | $3 \times 32 \times 32$ | $3 \times 64 \times 64$ | $3 \times 32 \times 32$ |
| # Classes/task | 5 | 5 | 2 |
| # Training samples/tasks | 2,250 | 2,250 | See Table 5 |
| # Validation Samples/tasks | 250 | 250 | See Table 5 |
| # Test samples/tasks | 500 | 250 | See Table 5 |

Table 5: 5-Datasets statistics (Saha et al., 2021b). For the datasets with monochromatic images, we replicate the image across all RGB channels so that size of each image becomes $3 \times 32 \times 32$. In split 5-Datasets, each dataset (in this table) is split into 5 tasks, each with 2 disjoint classes. 10% training data from each task is kept aside as validation sets.

|  | CIFAR-10 | MNIST | SVHN | Fashion MNIST | notMNIST |
|---|---|---|---|---|---|
| # Classes | 10 | 10 | 10 | 10 | 10 |
| # Training samples | 45,000 | 54,000 | 65,931 | 54,000 | 15,167 |
| # Validation Samples | 5,000 | 6,000 | 7,325 | 6,000 | 1,685 |
| # Test samples | 10,000 | 10,000 | 26,032 | 10,000 | 1,873 |

having 160 filters with $3 \times 3$ kernels, followed by two fully connected layers having 640 units each. For ImageNet-100 and miniImageNet experiments, we have used ResNet-18 model. This network consists of a front convolutional layer followed by 4 residual blocks each having four convolutional layers followed by a classifier layer. We used 40 filters in front convolutional layer and in the first residual block layers. For second, third and fourth residual blocks we used 80, 120 and 160 filters respectively. For ImageNet-100, in the front convolution layer, we used convolution with $7 \times 7$ kernel with stride 5. For miniImageNet, in the front convolution layer, we used convolution with $3 \times 3$ kernel with stride 2. For both of these cases, we used $2 \times 2$ average-pooling with stride 1 before the classifier layer. All the networks use ReLU in the hidden units and softmax with cross-entropy loss in the final layer.

### D.3 LIST OF HYPERPARAMETERS

A list of hyperparameters in our method and baseline approaches is given in Table 6. As in Gupta et al. (2020), hyperparameter for all the approaches are tuned by performing a grid-search using validation sets. For all the experiments, except split ImageNet-100, a batch size of 10 was used for training. In split ImageNet-100 experiments a batch size of 25 samples was used.

### D.4 BASELINE IMPLEMENTATIONS

For rehearsal-based methods - ER, GEM, A-GEM, MER and La-MAML, we used the implementation provided in La-MAML (Gupta et al., 2020). CLS-ER and DER++ are implemented by adapting the codes by Arani et al. (2022). EWC (Kirkpatrick et al., 2017), SI (Zenke et al., 2017) and NCL (Kao et al., 2021) are implemented adapting the codes from[1]. GPM (Saha et al., 2021b) and SGP (Saha & Roy, 2023) are implemented using the respective official open-sourced code repositories.

### D.5 AMPHIBIAN IMPLEMENTATION: SOFTWARE, HARDWARE AND CODE

We implemented Amphibian in `python (version 3.7.6)` with `pytorch (version 1.5.1)` and `torchvision (version 0.6.1)` libraries. We ran the codes on a single NVIDIA TITAN Xp GPU (CUDA `version 12.1`) and reported the results in the paper. To ensure reproducibility of these results, we attach the source codes of Amphibian with necessary instructions in **'Amphibian_Codes'** folder as the supplementary materials.

---

[1]https://github.com/GMvandeVen/continual-learning (MIT License)

Table 6: Hyperparameters grid considered for the baselines and Amphibian. The best values are given in parentheses. Here, 'lr' represents the learning rate. All the methods use SGD optimizer unless otherwise stated. The number of epochs in the OCL setup for all methods is 1. To maximally utilize the current batch of data in OCL, each method has a hyperparameter called glances (Gupta et al., 2020; Zhang et al., 2022) which indicates the number of gradient updates or meta-updates made on each of these batches. In the table we represent Split CIFAR-100 as 'cifar', Split TinyImagenet as 'tinyimg' and Split 5-Datasets as '5data'.

| Methods | Hyperparameters |
|---------|-----------------|
| ER | lr : 0.01 (5data), 0.03 (cifar), 0.1 (tinyimg); glances : 1 (5data), 10 (cifar, tinyimg) 
 memory size (data samples) : 100 (5data), 200 (cifar), 400 (tinyimg) |
| GEM | lr : 0.01 (5data), 0.03 (cifar, tinyimg); glances : 1 (5data), 2 (cifar, tinyimg) 
 memory size (data samples) : 100 (5data), 200 (cifar), 400 (tinyimg) |
| A-GEM | lr : 0.01 (tinyimg, 5data), 0.03 (cifar); glances : 1 (5data), 2 (cifar, tinyimg) 
 memory size (data samples) : 100 (5data), 200 (cifar), 400 (tinyimg) |
| MER | lr ($\alpha$) : 0.05 (5data), 0.1 (cifar, tinyimg) 
 lr ($\beta$) : 0.1 (cifar, tinyimg, 5data) 
 lr ($\gamma$) : 1.0 (cifar, tinyimg, 5data) 
 glances : 1 (5data), 10 (cifar, tinyimg) 
 memory size (data samples) : 100 (5data), 200 (cifar), 400 (tinyimg) |
| DER++ | lr : 0.01 (5data), 0.03 (cifar), 0.1 (tinyimg) ; glances : 1 (5data), 2 (tinyimg), 10 (cifar) 
 $\alpha$: 0.1 (tinyimg), 0.2 (cifar, 5data) ; $\beta$: 0.5 (cifar, tinyimg), 1.0 (5data) 
 memory size (data samples) : 100 (5data), 200 (cifar), 400 (tinyimg) |
| CLS-ER | lr : 0.01, 0.03, 0.05 (cifar, tinyimg, 5data) ; glances : 1 (5data), 2 (tinyimg), 10 (cifar) 
 $r_s$: $r_p$: 0.3 (5data), 0.5 (cifar), 0.9 (tinyimg) ; $r_p$: 0.5 (cifar), 0.8 (tinyimg), 1.0 (5data) 
 $\lambda$: 0.1 (cifar, tinyimg, 5data), 0.15 
 memory size (data samples) : 100 (5data), 200 (cifar), 400 (tinyimg) |
| La-MAML | $\alpha_0$ : 0.1 (cifar, tinyimg, 5data) 
 lr ($\eta$) : 0.25 (5data), 0.3 (cifar, tinyimg) 
 glances : 1 (5data), 2 (tinyimg), 10 (cifar) 
 memory size (data samples) : 100 (5data), 200 (cifar), 400 (tinyimg) |
| EWC | lr : 0.1 (cifar, tinyimg, 5data) 
 regularization coefficient, $\lambda$ : $1e^2$, $1e^3$ (cifar), $1e^4$, $2e^4$ (5data), $1e^5$ (tinyimg) 
 glances : 1 (5data), 2 (tinyimg), 5 (cifar) |
| SI | optimizer : Adam (cifar, tinyimg, 5data) 
 lr : $1e^{-3}$ (cifar, tinyimg, 5data) 
 regularization coefficient, $c$ : 1, 50 (tinyimag, 5data), 100 (cifar), 1000 
 glances : 1 (5data), 2 (tinyimg), 5 (cifar) |
| GPM | lr : 0.05, 0.1 (cifar, tinyimg, 5data) 
 $\epsilon_{th}$ : 0.96 (tinyimg), 0.975 (5data), 0.98 (cifar) 
 $\epsilon_{th}$ (increment/task) : 0.001 (cifar, tinyimg, 5data) 
 $n_s$ : 120 (cifar, tinyimg, 5data) 
 glances : 1 (5data), 2 (tinyimg), 5 (cifar) |
| SGP | lr : 0.05, 0.1 (cifar, tinyimg, 5data) 
 $\epsilon_{th}$ : 0.96 (tinyimg), 0.975 (5data), 0.98 (cifar) 
 $\epsilon_{th}$ (increment/task) : 0.001 (cifar, tinyimg, 5data) 
 scale coefficient ($\alpha$) : 1 (5data), 5 (cifar), 10 (tinyimg) 
 $n_s$ : 120 (cifar, tinyimg, 5data) 
 glances : 1 (5data), 2 (tinyimg), 5 (cifar) |
| NCL | lr : 0.05, 0.1 (cifar, 5data), 0.2 (tinyimg) 
 $p_w^{-2}$ : 2250 (cifar, tinyimg), 9000 (5data) 
 glances : 1 (5data), 2 (tinyimg), 5 (cifar) |
| Amphibian | $\lambda^0$ : 0.1, 0.25 (tinyimg), 0.5 (cifar, 5data) 
 lr ($\eta$) : 0.25, 0.5 (tinyimg), 1.0 (cifar, 5data) 
 glances : 1 (5data), 2 (tinyimg), 5 (cifar) |

# E ADDITIONAL RESULTS

## E.1 MEMORY OVERHEAD AND TRAINING TIME

In Figure 7(a) we show memory overhead comparisons during training for 5-Datasets tasks. In this case also, we observe that other baseline methods have orders of magnitude more memory overhead than Amphibian. Wall-clock training time comparisons among different methods for all three datasets are shown in Figure 7(b). Training times for all the tasks in the continual learning sequence for different experiments are measured on a single NVIDIA TITAN Xp GPU. As Amphibian uses inner-and outer-loop meta-learning steps, it requires more wall clock time for each model update during training compared to the other rehearsal-free baselines. However, other rehearsal-based meta-learners such as La-MAML and MER take up to ∼2.7× and ∼70× more training time than Amphibian.

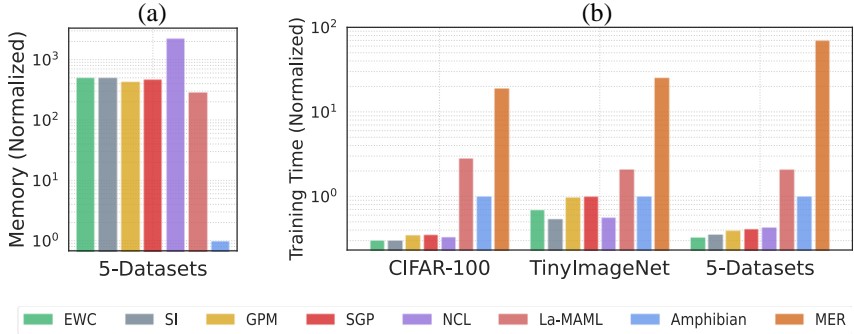

Figure 7: (a) Comparison of memory overhead (normalized by the Amphibian memory overhead) during training for 5-Datasets experiments. (b) Wall-clock training time comparisons for sequential training of all the tasks in different datasets. Normalized with respect to the time taken by Amphibian.

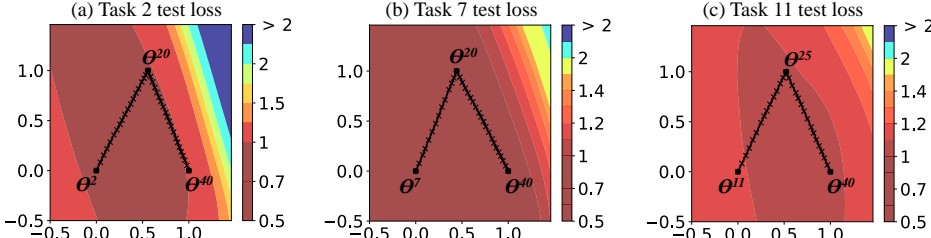

Figure 8: Dynamics of continual learning in Amphibian. Loss contours of (a) Task 2, (b) Task 7, and (c) Task 11 from split TinyImagenet dataset are plotted on 2D planes defined by parameters $(\theta^2, \theta^{20}, \theta^{40})$, $(\theta^7, \theta^{20}, \theta^{40})$ and $(\theta^{11}, \theta^{25}, \theta^{40})$ respectively. Black lines indicate learning trajectories.

## E.2 CONTINUAL LEARNING DYNAMICS IN AMPHIBIAN: A LOSS LANDSCAPE VIEW

**Loss Contour Plots.** We used visualization tools developed in Mirzadeh et al. (2020); Verwimp et al. (2021) to plot the loss contours (in Figure 6 and 8 ) on 2D planes defined by model parameters $(\theta^t)$. Each of these hyperplanes in the parameter space is defined by three points $\theta^1$, $\theta^2$ and $\theta^3$. Orthogonalizing $\theta^2 - \theta^1$ and $\theta^3 - \theta^1$ gives a two dimensional coordinate system with base vectors $u$ and $v$. The value at point $(x, y)$ is then calculated as the loss of a model with parameters $\theta^1 + u \cdot x + v \cdot y$. Please see the code/appendix in Mirzadeh et al. (2020) for more details.

**Continual Learning Dynamics in Amphibian.** In Figure 6 we showed that Amphibian updates incur minimum to no increase in losses of the past tasks for split CIFAR-100 tasks. Here, in Figure 8, we show the loss contours for three split TinyImagenet tasks. In Figure 8(a) we plot the loss contour of task 2 from TinyImagenet in a 2D plane. This figure shows when we sequentially learn from task 2 to task 40, along the entire learning trajectory ($\theta^2 \rightarrow \theta^{20} \rightarrow \theta^{40}$) loss of task 2 only increases

minimally from the initial point ($\theta^2$). A similar trend is also observed for other tasks. Here we show such trends for task 7 (Figure 8(b)) and task 11 (Figure 8(c)).

### E.3 Number of Gradient Scales in Amphibian

As we adopt low-dimensional gradient space representation (as discussed in Section 3) for gradients of weights/filters in the neural network, the number of learnable scales for gradient bases (in scale matrix, $\Lambda$) in Amphibian is very small compared to the size of the weights/filters. Table 7 shows the number of meta-learnable scales for each layer. Such a small number of scales explains the extremely low memory overhead of Amphibian during training as shown in Figure 4(c) and Figure 7(a).

Table 7: Number of meta-learnable gradient scales (in diagonal matrix, $\Lambda$) in each layer in Amphibian.

| Network | Layer | Size of Filters / Weights $(C_o \times C_i \times k \times k) / (m \times n)$ | Number of Scales (in Scale Matrix, $\Lambda$) |
|---|---|---|---|
| 5-layer Network (CIFAR-100) | Conv1 | $160 \times 3 \times 3 \times 3$ | 27 |
| | Conv2 | $160 \times 160 \times 3 \times 3$ | 1400 |
| | Conv3 | $160 \times 160 \times 3 \times 3$ | 1400 |
| | FC1 | $320 \times 2560$ | 2560 |
| | FC2 | $320 \times 320$ | 320 |
| 6-layer Network (TinyImagenet) | Conv1 | $160 \times 3 \times 3 \times 3$ | 27 |
| | Conv2 | $160 \times 160 \times 3 \times 3$ | 1400 |
| | Conv3 | $160 \times 160 \times 3 \times 3$ | 1400 |
| | Conv4 | $160 \times 160 \times 3 \times 3$ | 1400 |
| | FC1 | $640 \times 2560$ | 2560 |
| | FC2 | $640 \times 640$ | 640 |
| 5-layer Network (5-Datasets) | Conv1 | $200 \times 3 \times 3 \times 3$ | 27 |
| | Conv2 | $200 \times 200 \times 3 \times 3$ | 1800 |
| | Conv3 | $200 \times 200 \times 3 \times 3$ | 1800 |
| | FC1 | $400 \times 3200$ | 3200 |
| | FC2 | $400 \times 400$ | 400 |

