# OpenReview forum: "Amphibian: A Meta-Learner for Rehearsal-Free Fast Online Continual Learning"
_ICLR.cc/2024/Conference — Submitted to ICLR 2024_

### Official Review · Reviewer_kAVs · 2023-10-29

**Soundness:** 1 poor
**Presentation:** 2 fair
**Contribution:** 1 poor
**Rating:** 3
**Confidence:** 4

**Summary:**

This work proposes a continual learning approach called Amphibian, which is based on MAML-style model updates.
Unlike existing MAML-based approaches (e.g., Gupta et al., 2020), it does not rely on a replay buffer.
It assumes that the training examples are provided as a sequence of batches, each with dozens of examples.

The basic training scheme can be summarized as follows.
When a new batch comes in, it performs gradient descent for each example in the batch one by one (the inner-loop updates), producing a temporary model fitted to the batch.
The temporary model is then evaluated on the entire examples in the batch to yield the meta-loss.
Finally, the original model parameters are updated with the gradient w.r.t. the meta-loss (the outer-loop update), and the training proceeds to the next batch.

The main novelty of Amphibian is to introduce a gradient scaler $\lambda_i$ for each parameter $i$.
Whenever gradient descent is performed, this value is multiplied to the gradient, acting as a per-parameter learning rate.
This $\lambda_i$ is updated every batch by accumulating the products of the outer-loop gradient and the inner-loop gradient.

**Strengths:**

This paper dedicated significant effort to ensure reproducibility.
The appendix includes experimental details, and the code is provided in the supplementary material.

**Weaknesses:**

### Lack of Justification for the Method

Overall, the proposed method does not seem to have a solid theoretical basis.
The key idea of this work is to adjust the per-parameter learning rate with the cumulative sum of the products between the inner-loop gradients and the outer-loop gradients.
If the inner and outer gradients have the same sign in the current batch, the learning rate for the corresponding parameter is increased, and vice versa.
However, there is no justification for how such learning rate updates can be helpful to continual learning.

Interestingly, if we consider the case where the batch size is reduced to 1, this algorithm seems to become almost the opposite of EWC (Kirkpatrick et al., 2017).
In EWC, squared gradients are accumulated for each parameter, and the parameter becomes less flexible as the accumulated value grows.
In Amphibian with a batch size of 1, the inner gradient and the outer gradient are both computed with the only example in the batch.
Assuming the inner gradient does not incur drastic changes in the parameters, their product can be likened to the squared gradient in EWC.
However, Amphibian encourages the changes in the parameters with larger accumulated gradient products, which is the opposite of EWC.

### Confusing Notations

Starting from Eq. (7), $\ell_{in}$ and $\ell_{out}$ take only $\theta_0^j$ as input.
This ambiguates the meaning, especially for $\ell_{in}$.
According to the description under it, the inner loss $\ell_{in}$ is computed with a single example in a batch, but which example is it?
And why isn't there a summation of multiple $\ell_{in}$ from each example in the batch?

Similar confusion continues, even in the appendices.
For instance, $g_k$ in Eq. (14) and $g_{k'}$ in Eq. (16) seem to have the same definition with a different index, but their definitions are completely different.

I also do not see any utility in adopting the concept of gradient space.
Since the authors simply use $e_i$ as basis vectors, all the scaling is independently performed for each individual parameter.
Therefore, many equations can be simplified without introducing $e_i e_i^T$, which causes unnecessary confusion.
Similarly, the scale matrix $\Lambda$ can be simplified to per-parameter scale values.

Additionally, there is inconsistency in the subscripts for $\lambda$ and $e$.
The use of $i$ and $m$ is mixed in various instances, as seen in Equation (8) and (9).

I strongly recommend that the authors carefully restructure the overall notation in a systematic manner.


### Technically Incorrect Statements

#### Online Setting?

Although Amphibian is proposed as an online continual learning approach, one of its key assumptions is that the training examples are provided as a series of batches.
I think this is far from a truly online setting.
Generally, an online learning algorithm should be able to update a model meaningfully, even with a single example.
However, this is not the case for the proposed method.

#### Equivalence between Eq. (6) and (7)
The authors argue that Eq. (7) is *equivalent* to minimizing Eq. (6).
However, it seems to be an approximation, according to Appendix A.

---

In summary, I find limited value in the proposed method, and there is ample room for improvement, even in terms of its presentation. Consequently, I believe this paper does not meet the standards expected for an ICLR publication.

**Questions:**

How does Amphibian work in a fully online setting where each example is given individually, i.e., when $|\mathcal B_i| = 1$?

---

> ### Author Response · Authors · 2023-11-23
>
> We thank the reviewer for taking the time and review our paper. We provide responses to the reviewer’s questions/concerns below:
>
> According to the reviewer - “the key idea of this work is to adjust the per-parameter learning rate” and “all the scaling is independently performed for each individual parameter”. These statements are not true. As we clearly mentioned in the paper that our method learns a diagonal scale matrix at each layer and scale (adjust) the gradient directions with this matrix accordingly. To elaborate on this point, let’s assume we have a fully connected layer of a network having weight, W of size $m\times n$. Here, $m$ and $n$ are the numbers of output and input neurons respectively. There is a total of $mn$ parameters in this weight. Instead of learning $mn$ learning rate parameters, for this layer, we learn a diagonal scale matrix of size $n\times n$, thus only learning $n$ parameters. (Table 7 in the appendix lists the actual number of scale parameters learned by our methods in different experiments.) Alternatively, we can think of this as gradients of a group of $m$ parameters are scaled by a common factor ($\lambda_i$), and there are $n$ such groups in $W$ matrix. As we adopt the concept of low-dimensional gradient space (please see section 3 in the paper for details) such scaling is possible. This low-dimensional gradient representation requires the introduction of basis vectors, $e_i$. Because, in conventional representation, the gradient space would be $mn$ dimensional for the weight matrix. However, in our low-dimensional representation, it is $n$ dimensional. The $e_i$ vectors describe the basis of this space. The scales, $\lambda_i$ are associated with each of these vectors, which are learned/computed from the alignment history along the basis directions (Equation 8, 9).
>
> *On the justification of method*:
>
> We are using a bi-level (inner-loop and outer-loop)/meta optimization method for CL. In inner loop, we take $k$ number of gradient steps, where each step uses a (non-overlapping) subset of data from the current batch $\mathcal{B_i}$. These inner loop updates intuitively simulate a ‘mini continual’ learning scenario. At the end point of inner-loop update the outer-loop loss on the whole batch $\mathcal{B_i}$ is computed. This loss indicates how well the current samples are learned and how much forgetting is there due to such (mini continual) learning. The outer-loop update then adjusts the weight to minimize forgetting and ensure good learning on the given batch. Now, to have learning synergy among different sequential batches, we use scale matrix that stores directional gradient alignment information. The direction along which samples in the online sequence have positive (negative) cumulative gradient alignment, model updates along those directions are encouraged (prevented). With extensive experiments with diverse datasets and architectures, we have shown the effectiveness of this method. Also, we show that loss on the previous tasks has minimal to no increase due to such updates which further justifies our method for CL.
>
> Now, $\mathcal{|B_i|}=1$ ($k=1$) is not an ideal scenario for an meta-learning method like ours. This would mean there is no mini-CL sequence to observe in inner loop. Thus in the outer-loop no meaningful gradient information (for CL) will be generated for model update. For the same reason, the scale update would not be useful, as we won’t be able to measure and store any cross-sample gradient alignment information (Eq 8,9), which plays a key role in Amphibian. This also explains the alleged inconsistency between functionality of EWC and Amphibian. In  $\mathcal{|B_i|}>1$ ($k>1$) case, however, Amphibian produced expected CL performance. In Section 6.2 (and in Figure 5(c)-(d)) we show how Amphibian and EWC become functionally equivalent methods for CL with an added constraint with hyperparameter $\beta$.
>
> To enable Amphibian work with  $\mathcal{|B_i|}=1$, a temporary memory buffer of size $B$ would be needed. Typically, $B$ would be of size of 5 to 10 samples. For each incoming example, the model would take an inner gradient step and that example will be stored in that buffer. Once the buffer is full, outer gradient will be computed on the entire buffer data and then the buffer will be cleared out immediately. This is how Amphibian would work in the reviewer-suggested truly/fully online setup.
>
> *On the confusing notations*: We thank the reviewer for pointing this out. We will correct the typos (e.g. $i$ vs $m$) and restructure the notations (wherever applicable) in the revised manuscript.
>
> *On the technically incorrect statements*: (1) We used the definition of online setting as per literature (La-MAML, A-GEM, ASER ) where training examples are provided as a series of batches. (2) Yes, Eq. (6) and 7 are equivalent under approximations (Appendix A). We will highlight this point in the manuscript.

---

> > ### Comment · Reviewer_kAVs · 2023-12-03
> >
> > I appreciate the authors for providing a detailed response.
> > Unfortunately, however, my concerns are not fully resolved.
> >
> > The authors' explanation regarding the per-parameter learning rate appears inconsistent with Eq. (4).
> > If parameter $\theta$ in Eq. (4) represents an $mn$-dimensional vector of the parameters, $\Lambda$ should be a diagonal matrix with a dimension of $mn \times mn$.
> > I believe the description and notations should be improved significantly.
> >
> > The justification for the method seems somewhat vague and lacks a rigorous argument. Moreover, the inability to handle small batch sizes could present a significant drawback to this approach.
> >
> > Consequently, I will maintain my initial score.

---

### Official Review · Reviewer_esZ7 · 2023-10-30

**Soundness:** 2 fair
**Presentation:** 3 good
**Contribution:** 2 fair
**Rating:** 5
**Confidence:** 4

**Summary:**

This paper presents a rehearsal-free continual learning algorithm, based on La-MAML. It employs bi-level optimization to learn a diagonal scale matrix in each layer, aiming to prevent catastrophic forgetting. Comprehensive experiments and analyses demonstrate its superior experimental performance. However, there may be an unfair comparison setting that needs clarification.

**Strengths:**

+ The paper focuses on Task-IL incremental learning and significantly improves performance in the realm of rehearsal-free methods.
+ I commend the authors for conducting comprehensive analysis experiments to evaluate the proposed Amphibian. These include Task Learning Efficiency, visualization of the loss landscape, and a comparison of few-shot forward transfer.

**Weaknesses:**

+ I have concerns about the fairness of the comparable online setting. In both La-MAML and the code provided in your appendix, you have the hyper-parameter 'self.glances', which allows your online training batch to be optimized multiple times. While it's understandable that the early CL work La-MAML adopts this 'single-pass' setting due to the lack of clear definitions for online and offline CL settings, if you're adopting the online CL setting, you need to clearly highlight the differences between your experimental setting and the standard online CL setting where each example can only be seen once. Furthermore, you should provide results of other comparable methods under this setting or set your hyperparameter 'self.glances' to 1 for a fair comparison.
+ La-MAML, as the most important baseline, also learns the learning rate through bi-level optimization, similar to your learned diagonal scaled matrix. Despite the results provided in Table 3, I'm still unclear if the learned diagonal scaled matrix truly outperforms the learned learning rate for each parameter of La-MAML. The differences between La-MAML and the proposed Amphibian are:
    - La-MAML uses samples from the memory buffer, while Amphibian does not.
    - Both La-MAML and Amphibian apply the ReLU operation on the learned learning rate or the diagonal scaled matrix. However, La-MAML only applies this ReLU operation during the outer loop, while Amphibian uses it in both the inner and outer loops. Existing research [1] shows that using the ReLU operation on the learning rate during both inner-loop and outer-loop can effectively improve performance. So it is unclear if your performance gains lies in this different operation.
In Table 3, you only show the ablation study on the first point. Therefore, it doesn't convince me that the learned diagonal scaled matrix is truly superior to the learning rate learned by La-MAML.
    ```
    Reference: [1] Learning where to learn: Gradient sparsity in meta and continual learning.  NeurIPS, 2021
    ```
+ In my view, the learned diagonal scaled matrix is equivalent to learning the important weights for the current task. However, like EWC, it learns the important weights (i.e., the Fisher information matrix) for each task and suppresses the model’s updates in these directions. I'm still unsure how the timely learned diagonal matrix can prevent catastrophic forgetting of previous tasks. I believe the authors need to provide more explanations. Is the proposed method, Amphibian, only applicable in the relatively simple Task-IL setting? Providing the Class-IL online CL performance could be much more convincing.

**Questions:**

+ If you're adopting the online CL setting, you need to clearly highlight the differences between your experimental setting and the standard online CL setting where each example can only be seen once.
+ It's unclear how the timely learned diagonal matrix can prevent catastrophic forgetting of previous tasks.
+ Is the proposed method, Amphibian, only applicable in the relatively simple Task-IL setting? Could you provide the Class-IL online CL performance?

Please see the weakness section for more details.

---

> ### Author Response · Authors · 2023-11-23
>
> We thank the reviewer for taking the time and review our paper. We provide responses to the reviewer’s questions/concerns below:
>
> *On the Weaknesses*:
>
> 1. As per La-MAML we used the online (single epoch) learning setup where there is a glance hyperparameter. To have a fair comparison, in Table 1 we have reported results for all the other baselines in the same setup where multiple glances are allowed. Table 6 in the Appendix lists all the hyperparameters for these baselines, which contains the glance values as well.
>
> 2. In their implementation, La-MAML used ReLU operation in both inner and outer loops (Please see official La-MAML implementation [1] lines 55 and 99). In Table 1 we showed results for La-MAML (with rehearsal) and in Table 3 La-MAML (without rehearsal). Comparing these results we see that La-MAML with replay performs significantly better in terms of accuracy and forgetting mitigation. Since our method does not use any rehearsal and outperforms La-MAML (with rehearsal), learned diagonal matrices (unique component of our method) play a major role in preventing catastrophic forgetting.
>
> 3. The proposed method can also work in Class-IL online setup and the results are given already in the manuscript (in Figure 4, section 6.1).
>
> [1] https://github.com/montrealrobotics/La-MAML/blob/main/model/lamaml.py

---

> > ### Comment · Reviewer_esZ7 · 2023-12-02
> >
> > Thanks for the authors' response. I would like to keep my score since I don't think "multiple glances" in online CL is that realistic.

---

### Official Review · Reviewer_ebWh · 2023-11-08

**Soundness:** 3 good
**Presentation:** 2 fair
**Contribution:** 2 fair
**Rating:** 3
**Confidence:** 3

**Summary:**

This paper presents a new algorithm tailored for the online continual learning paradigm, which operates without the need for rehearsal. It also offers a theoretical analysis of the approach. The method is characterized by its learning of a layer-wise diagonal scale matrix that captures the historical trajectory of gradient updates. The paper conducts a comparative evaluation of the proposed algorithm against established methods in the field of continual learning and provides a detailed analysis of the outcomes.

**Strengths:**

1. The experimental section of this article is quite comprehensive and theoretical analysis are provided.

2. The authors design a novel rehearsal-free algorithm for continual learning, which achieves commendable results.

**Weaknesses:**

1. This work just adds an adaptive diagonal scale matrix in each layer, which seems trivial. The contribution is somewhat limited.

2. The authors allocate a substantial portion to the analysis of experimental results. Although the necessity of the experiment is clear, the analysis could benefit from being more concise to avoid redundancy.

3. The presentation could be improved to get better readability.

**Questions:**

1. Could you please provide a more detailed explanation of how the proposed method differs from La-MAML?

2. Could you explain the rationale behind constraining the matrix to a scale matrix?

---

> ### Author Response · Authors · 2023-11-23
>
> We thank the reviewer for taking the time and review our paper. We provide responses to the reviewer’s questions/concerns below:
>
> *Answer to Q1*: The main differences between our method (Amphibian) and La-MAML are: (1) La-MAML uses a replay buffer for storing old data, whereas Amphibian does not use any replay buffer. Thus La-MAML is a rehearsal-based method whereas Amphibian is a rehearsal-free method. (2) LA-MAML learns per parameter learning rate using data from both current and past tasks, whereas Amphibian learns layer-wise diagonal scale matrix using only the current batch data. Thus, the number of extra learnable parameters (memory overhead) in La-MAML is very high compared to Amphibian (Please see Figure 4(c)). (c) Amphibian minimizes a meta-objective that encourages alignments of gradients among given data samples along selected basis directions in the gradient space (Eq 7), La-MAML optimizes a different meta-objective.
>
> *Answer to Q2*: We adopted a low-dimensional gradient space representation in each layer of the network (please see section 3 for details). For each basis describing this space, we wanted to assign a scale by which incoming gradients will be modified during CL training. For this purpose, we introduced a diagonal scale matrix (in Eq. 4,5).

---

> > ### Comment · Reviewer_ebWh · 2023-12-02
> >
> > Thank you for your response. After reading the author's rebuttal and the opinions of other reviewers, I decide to maintain the original rate unchanged.

---

### Meta-Review · Area_Chair_mvvH · 2023-12-02

**Metareview:**

**Summary:**

This paper introduces a novel continual learning approach named Amphibian, which is based on MAML. Different from existing methods, Amphibian eliminates the need for rehearsal, a common requirement in MAML-based algorithms. Its key feature is the implementation of a learned diagonal scale matrix in each layer, designed to mitigate catastrophic forgetting. The paper presents a thorough evaluation of the proposed method against existing algorithms, and offers a detailed analysis of the results.

**Strengths:**

1. The experimental section of this article is comprehensive and theoretical analysis is provided.
2. Amphibian demonstrates improved performance in the domain of rehearsal-free continual learning methods, which is a notable advancement.

**Weaknesses:**
1. The primary innovation, adding an adaptive diagonal scale matrix to each layer, is somewhat incremental. This approach is similar to learning important weights for the current task.
2. It's unclear why the learned diagonal matrix can prevent catastrophic forgetting of previous tasks. It requires more convincing explanations.

**Justification For Why Not Higher Score:**

1. The fairness of the experimental comparison setup is questionable, potentially impacting the validity of the results.
2. The paper lacks an analysis of how the diagonal matrix prevents catastrophic forgetting, a critical aspect of continual learning.
3. The paper falls short in convincingly demonstrating that Amphibian really outperforms La-MAML.

**Justification For Why Not Lower Score:**

N/A

---

### Decision · Program_Chairs · 2024-01-16

Reject